

# A global analysis of water storage variations from remotely sensed soil moisture and daily satellite gravimetry

Daniel Blank[1], Annette Eicker[1], Laura Jensen[1], and Andreas Güntner[2,3]

[1]HafenCity University Hamburg, Hamburg, Germany
[2]Helmholtz Centre Potsdam GFZ German Research Centre for Geosciences, Potsdam, Germany
[3]University of Potsdam, Institute of Environmental Sciences and Geography, Potsdam, Germany

**Correspondence:** Daniel Blank (daniel.blank@hcu-hamburg.de)

**Abstract.** Water storage changes in the soil can be observed on a global scale with different types of satellite remote sensing. While active or passive microwave sensors are limited to the upper few centimeters of the soil, satellite gravimetry can detect changes of terrestrial water storage (TWS) in an integrative way but it cannot distinguish between storage variations in different compartments or soil depths. Jointly analyzing both data types promises novel insights into the dynamics of subsurface water storage and of related hydrological processes. In this study, we investigate the global relationship of (1) several satellite soil moisture products and (2) non-standard daily TWS data from the GRACE and GRACE-FO satellite gravimetry missions on different time scales. The six soil moisture products analyzed in this study differ in post-processing and the considered soil depth. Level-3 surface soil moisture data sets of SMAP and SMOS are compared to post-processed Level-4 data products (surface and root zone soil moisture) and the ESA CCI multi-satellite product. On a common global 1 degree grid, we decompose all TWS and soil moisture data into seasonal to sub-monthly signal components and compare their spatial patterns and temporal variability. We find larger correlations between TWS and soil moisture for soil moisture products with deeper integration depths (root zone vs. surface layer) and for Level-4 data products. Even for high-pass filtered sub-monthly variations, significant correlations of up to 0.6 can be found in regions with large high-frequency storage variability. A time-shift analysis of TWS versus soil moisture data reveals the differences in water storage dynamics with integration depth.

## 1 Introduction

Freshwater stored on the continents sustains life on Earth and is a key variable in the global cycles of water, energy and matter. Among the different continental storage compartments that make up terrestrial (or total) water storage (TWS), such as glaciers and ice caps, surface water bodies and groundwater, soil moisture (SM) plays a particularly important role at the soil-vegetation-atmosphere interface. Recognizing its important control on numerous processes in the climate system, SM and TWS have been declared as Essential Climate Variables (Dorigo et al., 2021a). SM is defined as the water contained in the unsaturated soil zone, i.e. the zone above the groundwater table that is not completely filled with water. Even though SM only accounts for 0.05 % of the total freshwater resources on Earth (Shiklomanov, 1993), SM is fundamental in providing the water supply for the Earth's vegetation cover and for ecosystems in the critical zone. In spite of its small absolute volume, SM can make a large contribution to TWS variations (e.g., Güntner et al. (2007)). SM is directly influenced by water fluxes



at the land surface such as precipitation, snow melt and evapotranspiration and plays a decisive role on how the water input is
distributed among root water uptake, groundwater recharge or runoff, for instance, i.e., how water fluxes are partitioned between
different storage compartments. While near-surface SM usually exhibits high fluctuations on short time scales due to its direct
exposure to the hydro-meteorological forcing, temporal SM variations tend to be smoother and delayed with increasing soil
depth (e.g. Xu et al. (2021b)) while the degree of coupling between near-surface, deeper or depth-integrated water storage

may vary considerably with the site conditions (e.g.,Carranza et al. (2018)). Overall, jointly analyzing both SM and TWS
data sets may reveal better insights into the hydrological dynamics and the processes that govern water storage changes in the
subsurface. Monitoring SM and TWS is thus crucial for understanding variations and changes in the global water cycle. In spite
of considerable efforts in collecting in-situ SM observations at the global scale (Dorigo et al., 2021b), a global coverage that
includes even remote regions of the globe is only possible by means of satellites. Even more for TWS monitoring, only very

few in-situ observations exist (e.g., Güntner et al. (2017)) while the large-scale and global coverage can be achieved by remote
sensing. Two different types of satellite observations are sensitive to changes of water in the subsurface: (1) Active or passive
microwave remote sensing can observe SM in the top few centimeters of the soil exploiting the fact that the dielectric constant
of the soil changes with varying soil water content. Several dedicated instruments are currently in operation on different satellite
missions (e.g. SMOS, SMAP, ASCAT or AMSR-2). Also, first results were obtained based on reflected signals of Navigation

Satellite Systems received in space (e.g., Camps et al. (2016), Chew and Small (2018), Kim and Lakshmi (2018)). (2) Due to
the fact that any redistribution of water mass on or above the Earth's surface leads to variations of the Earth's gravity field,
satellite gravimetry can relate changes in the gravitational acceleration acting on a satellite to variations TWS, which includes
SM. The twin satellite mission Gravity Recovery and Climate Experiment (GRACE, Tapley et al. (2004)) and its successor
mission GRACE Follow-On (GRACE-FO, Landerer et al. (2020)) have been observing gravity field changes since 2002.

The two types of satellite observations (remotely sensed surface soil moisture (SSM) and TWS from satellite gravimetry)
both have their individual advantages and drawbacks. Satellite gravimetry is sensitive to all parts of TWS on and underneath
the Earth's surface, but cannot distinguish between water mass changes in individual water compartments (e.g. snow cover,
groundwater, SM, surface water). The separation of the integrative signal to study individual compartments, e.g. SM variations,
is challenging (Schmeer et al., 2012). Furthermore, the low spatial resolution of the GRACE data of a few 100km makes the

analysis of local phenomena difficult. SM remote sensing provides a much higher spatial resolution (20-40km), but observes
SM only in the top few centimeters (∼2cm) of the soil (Escorihuela et al., 2010). Also, measuring SM with microwave satellites
is problematic in regions such as tropical rain forests due to dense vegetation or in arctic regions because of snow cover or frozen
ground (Karthikeyan et al., 2017). Whether (and how) the satellite-based surface SM can be used to empirically extrapolate
wetness conditions into deeper soil layers and thus give evidence of large-scale contributions to the global water cycle is an

open research question (De Lannoy and Reichle, 2016). Existing vertical extrapolation or depth-scaling algorithms exhibit large
discrepancies and their validation is difficult (Zhang et al., 2017). The comparison of the extrapolated root-zone SM (RZSM)
dynamics against the integrative observations of TWS variations can be a valuable means of evaluating the depth-scaling
approaches, in particular in areas where TWS is dominated by water storage variations in the unsaturated zone.





However, the temporal resolution of standard GRACE data of one month is too low to record any fast water mass changes
in the upper soil layers. Recent developments in GRACE data processing (Kvas et al., 2019) have enabled the computation
of daily gravity fields with increased accuracy. Such daily gravity data were successfully used to study high-frequency wind-
driven sea-level changes (Bonin and Chambers, 2011), short-term transport variations of the Antarctic Circumpolar Current
(Bergmann and Dobslaw, 2012), the characteristics of major flood events (Gouweleeuw et al., 2018), and high-frequency
atmospheric fluxes (Eicker et al., 2020). Furthermore, the daily gravity-based TWS data appear particularly promising to
capture SM variations at short time scales, but have not been used for this purpose yet.

While a large number of studies have examined TWS and SM individually, joint (global) analyses of the two observation
types are largely missing. Only Abelen et al. (2015) and Abelen (2016) have provided first comparisons between two satellite
SM products and TWS, but only on monthly and not on daily time scales. Besides the direct data comparison, the joint assim-
ilation of both observables into hydrological models currently is an emerging field (e.g. Tian et al. (2019); Tangdamrongsub
et al. (2020)).

Given the potential value of combining SM and TWS observations for understanding water cycle dynamics as outlined above,
the aim of this study is to investigate the global relationship between (1) non-standard daily TWS data from GRACE(-FO) and
(2) several satellite SM products on different time scales to derive novel information content on Earth system dynamics. An
overview of the data sets and their characteristics is shown in Fig. 1. The six SM products analyzed in this study can be
categorized based on their degree of post-processing and their integration depth into the soil. Original Level-3 surface SM data
sets of SMAP (Entekhabi et al. 2010) and SMOS (Kerr et al. 2010) are compared to post-processed Level-4 data products
(surface and root zone SM) and a multi-satellite product provided by the ESA Climate Change Initiative (CCI) (Gruber et al.
2019, Dorigo et al. 2017). While strong correlations of the two data types in the dominant seasonal cycle are to be expected,
finding such correlations on shorter (down to sub-monthly and daily) time scales would be a first-time evidence that satellite
gravimetry can indeed observe such fast-changing SM signals, opening new opportunities of applications of this observation
technology.





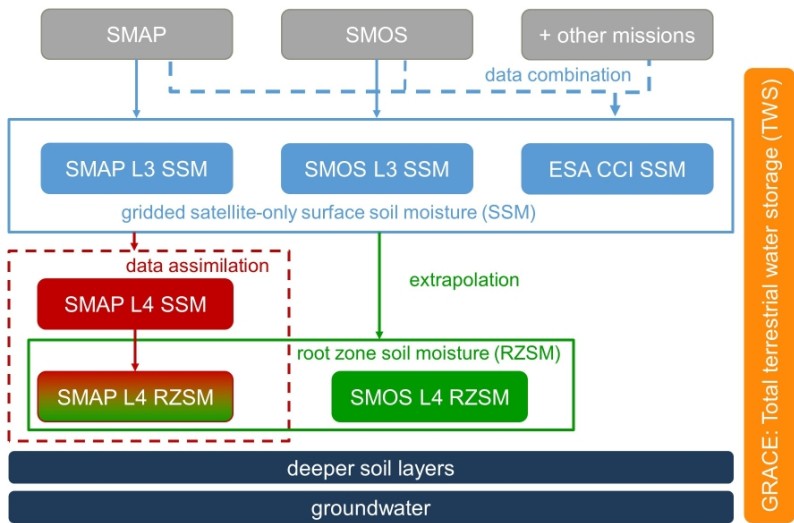

**Figure 1.** An overview of the soil moisture data sets used and their characteristics.

The paper is organized as follows: In Section 2 we describe the satellite data products used for the comparison of the different data sets (SM and TWS) followed by information on the applied data processing in Section 3. The results are presented in Section 4, first for the full signal and afterwards for sub-monthly time scales isolated by high-pass filtering. In both cases, we first illustrate signal characteristics using an exemplary time series for a specific grid cell followed by global maps of correlation coefficients and relative time shifts to show the (dis-)agreement between the data types.

## 2 Data sets

### 2.1 GRACE and GRACE-FO data

To investigate fast temporal water storage changes, we use daily gravity field solutions of the ITSG-Grace2018 model (Kvas et al., 2019) converted to global time series of daily TWS anomalies. Compared to the standard monthly solutions, the limited satellite ground track coverage during one day does not allow for a stable global gravity field inversion, thus additional information has to be introduced. The processing of the daily solutions is, therefore, carried out using a Kalman smoothing approach (similar to Kurtenbach et al. 2012), which introduces statistical information on the expected evolution of the gravity field over time in its process model.

Daily gravity field data are given in the form of spherical harmonic coefficients (so-called Level 2 products) of the gravitational potential up to a degree of $n = 40$ corresponding to a spatial resolution of 500 km. During the data processing, temporally high-frequency mass variations caused by tides (ocean, Earth and pole tides), as well as non-tidal atmospheric and ocean mass variations are removed by subtracting the output of geophysical background models from the observations (de-aliasing, Dob-





slaw et al. (2017)). Post-processing steps account for the effect of geocenter motion (adding the degree-1 harmonic coefficients
given by Sun et al. (2016) on the basis of Swenson et al. (2008)), replace the $c_{20}$ coefficient based on a time series from Satel-
lite Laser Ranging (Cheng and Ries, 2017) and subtract the influence of glacial isostatic adjustment (GIA) using the ICE6G-D
model (Peltier et al., 2017). No extra spatial filtering is needed, because the Kalman smoother effectively suppresses spatially
correlated noise. After these processing steps the resulting gravity field models are assumed to primarily contain water mass
changes above and below the Earth surface and can be converted to equivalent water heights on a global geographical grid of
$1° \times 1°$ according to

$$TWS(\lambda, \vartheta) = \frac{M}{4\pi R^2 \rho_w} \sum_{n=1}^{n_{max}} \sum_{m=-n}^{n} \frac{(2n+1)}{(1+k'_n)} c_{nm} \Upsilon(\lambda, \vartheta) \tag{1}$$

where $\lambda$ and $\vartheta$ symbolize the spherical coordinates, $M$ and $R$ are the mass and the radius of the Earth, $\rho_w$ is the density
of water ($\frac{1000\,kg}{m^3}$), $k'_n$ denote the Load Love Numbers (Lambeck, 1988), $c_{nm}$ are the spherical harmonic coefficients of the
gravitational potential and $\Upsilon(\lambda, \vartheta)$ are the surface spherical harmonic functions. The degree and order of the spherical har-
monic functions is denoted by $n$ and $m$. For a reasonable overlap with all satellite-based SM products, we used the time period
from April 2015 to December 2021 for our study, excluding the time span between the end of the mission GRACE (August
2017) and the start of the successor mission GRACE-FO (July 2018). Even though the Kalman smoother output provides a
continuous daily time series without data gaps for the mission time periods, all days with insufficient GRACE observations
were excluded from our analysis, i.e., days with an observation count of less than 10.000 observations per day as given on the
website of the ITSG-Grace2018 product. On these days, the daily solutions are mainly informed by the process model of the
Kalman filter and thus tend towards an a-priori mean trend and annual signal.

## 2.2 Soil Moisture Data Sets

Active or passive microwave remote sensing can observe SM in the top few centimeters of the soil exploiting the fact that the
dielectric constant of the soil changes with varying soil water content. Several dedicated instruments are currently in operation
on different satellite missions. Active microwave sensors (radars) transmit an electromagnetic pulse to the Earth and measure
the pulse's backscattered energy from the surface of the Earth, whereas passive microwave sensors (Radiometers) observe
radiation naturally emitted by the Earth's soil, which is expressed as brightness temperature (Robinson et al., 2008). The
observed parameters (backscattered energy and brightness temperature) of both techniques depend on the dielectric constant
of the soil, which allows the measurement of SM.

In our study we use satellite-derived SSM products from the missions SMOS, SMAP and from the combination data product
ESA CCI, as well as RZSM products from SMOS and SMAP. The overlapping time span of all missions between April 2015
(start of the SMAP mission) and December 2021 was selected. In addition, it should be mentioned that the orbit direction
(ascending/descending orbit) differs across the Level 3 products and therefore also the overpass time. In this study, the early
morning overpass is chosen for the Level 3 satellite SM products. This is suggested for passive measurement methods, as the





temperature difference between the soil surface and the vegetation canopy in the morning and night, as well as the thermal difference between various types of land cover within a pixel is reduced, resulting in a minimization of SM retrieval errors and a better reliability (Owe et al. 2008, Entekhabi et al. 2014, Lei et al. 2015, Montzka et al. 2017).

### 2.2.1 SMOS

The Soil Moisture Ocean Salinity (SMOS, Kerr et al. 2010) satellite was implemented by ESA (European Space Agency) as part of the Earth Explorer missions and launched in November 2009. The satellite operates in a sun synchronous orbit and the ascending orbit overpasses the equator at 6 am local time. SM is observed by using an L-band radiometer, which receives the radiation emitted by the Earth's surface and measures the brightness temperature. This technique allows observations in the first few centimeters of soil (SSM). SMOS needs less than 3 days to revisit the same area with a maximum spatial resolution

of around 40-50 km.

In this study, we use the daily Level 3 (v3.0) SSM product of the Centre Aval de Traitement des Données SMOS (CATDS), operated by the Centre National d'Etudes Spatiales (CNES). The L3 data contain all collected SM data for each day on a global grid with a spatial resolution of 25 km x 25 km (Al Bitar et al., 2017). Additionally the CATDS Level 4 RZSM product is used, which propagates the L3 SSM data set (0-5 cm) into the underlying soil (5-40 cm) using an exponential filter. Then, from the

40 cm layer to the root zone layer (up to 1m soil depth), a budget model based on a linearized Richards Equation formulation is adopted to compute the water content (Al Bitar et al., 2013). The RZSM is a weighted average of the two layers expressed in $m^3/m^3$. Starting in February 2020 the SMOS L4 data set has been processed using a new algorithm. The method uses the SSM product from SMOS to calculate the SM of the root zone (1 meter depth) based on a modified formulation of a recursive exponential filter, while considering soil properties and an optional implementation of transpiration (Al Bitar and Mahmoodi,

2020). As the data from 2015 to 2020 have not yet been reprocessed using the new algorithm, we decided to concatenate the two time series in order to adhere to the general comparison period of April 2015 to December 2021. An offset between the two differently processed time spans was calculated from the 14 days overlap period from 31.01.2020 till 13.02.2020 and removed from the February 2020 till December 2021 data to achieve a seamless transition with the previous data (April 2015 till January 2020). The L4 RZSM data set is provided in a global grid with a spatial resolution of 25 km x 25 km.


### 2.2.2 SMAP

The Soil Moisture Active Passive (SMAP, Entekhabi et al. 2010) satellite was launched in January 2015 by the National Aeronautics and Space Administration (NASA) to observe SSM with an L-band active radar and passive radiometer. After a few months the active system failed and since then only the passive system is operational, which observes the brightness

temperature of the earth. Like the SMOS satellite, SMAP operates in a sun synchronous orbit but in contrast to SMOS the overpass of the equator at 6 am local time occurs on the descending orbit. SMAP needs a maximum of 3 days to revisit the same area and measures the brightness temperature with a spatial resolution of around 40 km.





In this study Level 3 and Level 4 products from SMAP are used. The daily Level 3 data (v8.0, ONeill et al. (2021)) contain all SSM retrievals for an entire day mapped to a global grid with a spatial resolution of 36 km. In contrast, the Level 4 prod-

ucts are derived by assimilating SMAP surface brightness temperatures observations into the Godard Earth Observing Model System (GEOS-5, v5) catchment land surface model. The land surface model is driven by observation-based surface meteorological forcing data, including precipitation, and represents essential land surface processes, such as the vertical movement of water in the soil between the surface and the root zone. Finally, the assimilation system interpolates and extrapolates SMAP observations in time and space using the land model, which gives estimates for the SSM (5 cm depth) as well as the RZSM (up

to 1 m depth) provided with a temporal resolution of 3 hours and a 9 km spatial resolution. Both Level 4 data sets (SSM and RZSM v6.0, Reichle et al. (2021)), available from the website of NSIDC, are used in our analysis. To evaluate all products at the same daily scale the SMAP L4 products are re-sampled to daily data by taking the average of all observations for one day (8 observations per day for a 3-hour temporal resolution).

### 2.2.3 ESA CCI

The ESA CCI SM data (Gruber et al. 2019, Dorigo et al. 2017) set is provided as part of ESA's Climate Change Initiative (CCI). The ESA CCI SM product is based on harmonizing and merging SM retrievals from multiple satellites into a combined daily product. Three different data sets are provided: an active-microwave-based-only product, a passive-microwave-based only product and a combined active and passive SM product.

We select the combined active and passive SSM product (v7.1) for our analysis. It provides daily SM observations with a spatial resolution of 0.25° and is available from the ESA data archive. This product includes SM retrievals from the active satellites AMI-WS ERS-1/2 SCAT and METOP-A/B ASCAT and from the passive satellites NIMBUS 7 SMMR, DMSP SSM/I, TRMM TMI, Aqua AMSR-E, Coriolis Windsat, SMOS, GCOM AMSR2 and SMAP. Overall, this product covers SM data from 1978 to the present. It should be noted that tropical rainforest areas are completely masked out because of the strong

signal scattering in the microwave observations caused by vegetation (Ulaby et al., 2014).

## 3 Data processing

The comparison between satellite gravimetry and the SM products is carried out on grid cell level, therefore the SM products are harmonized to the same 1° x 1° geographical grid used in Eq.(1) to compute TWS from the GRACE & GRACE-FO data. Downsampling was performed using a first order conservative remapping function, which leads to a lower spatial resolution

for the various SM data sets. Despite a remaining difference in the spatial signal content between the frequency-limited TWS and the gridded SM data no further downsampling was performed in order to preserve the characteristics of the SM time series.

The linear trend of the time series is removed before examining the agreement between SM and TWS. To isolate sub-monthly variations, a third-order Butterworth high-pass filter with a cutoff-frequency of 30 days is applied in forward and backward





direction (to avoid a phase shift). This filter conserves the phase, but removes signals with periods longer than 30 days which

dominate the time series. The computation of the high-pass filtered signal is presented in further detail in the Appendix B.

Since a direct comparison of the absolute values of SM and TWS is not possible due to the different integration depths and units of the respective data sets, we analyze their relationship using Pearson's pairwise correlation coefficient $\rho_{xy}$. Possible time lags between TWS and SM are determined using cross-correlation analysis, which indicates the time shift for which two signals best agree with each other. The Appendix A1 provides further information on these metrics. The data gaps in the TWS

data set (see Section 2.1) were also excluded from the SM time series prior to comparison. No further temporal masking, e.g. based on quality flags, was performed for the SM data.

### 3.1 Spatial mask

The type of land cover can lead to limitations for observing SM. In densely vegetated areas such as tropical rainforests, the observed emissivity by passive satellites or the backscattered energy by active satellites is primarily caused by the vegetation

(Ulaby et al. 2014, Owe et al. 2001). In deserts, the SM signal may be unreliable as the variability of water in the upper soil is low (Dorigo et al., 2010). The problem of reduced sensitivity in deserts also accounts for the satellite gravimetry observations from GRACE & GRACE-FO as their signal-to-noise ratio is low and the noise floor of GRACE strongly dominates the time series. Other surface characteristics that limit measurements of satellite SM are snow cover and frozen soil since the dielectric constant of snow and frozen water varies significantly from the one of liquid water (Wagner, 1998).

Therefore, three spatial masks have been defined to identify problematic regions for observing SM: dense vegetation, regions with snow and frozen ground throughout large parts of the year, and areas with low SM (e.g. deserts), as seen in Fig. 2. To indicate dense vegetation, we use in our study the tropical rain forest mask that is also applied in the ESA CCI data set. Also for regions with a large fraction of snow cover and frozen soil, the same method as the ESA CCI mask is used. It uses soil temperature ($T_S$) and snow water equivalent (SWE) estimates from GLDAS NOAH to flag satellite observations that were

taken under the conditions of frozen soil ($T_S < 0°$ C) and snow cover (SWE $> 0$ mm) (Gruber et al., 2019). Grid cells in which more than 40 % of all observations are influenced by these conditions were included in the mask. We classify dry (desert) regions based on the SM signal variability, i.e., the root mean square (RMS) of the daily SM time series in each grid cell. As a threshold value for low variability we use twice the mean RMS of a test region in the Sahara desert where no major day-to-day SM variations can be expected. For defining the mask with this criterion, we used the SMOS L3 data product. Using the the

other SM data products led to very similar results. In some regions, the low SM variability mask overlaps with the snow cover and frozen ground mask. The low SM variability areas are excluded from further analyses because no discernible signals of both SM and TWS can be expected. As we want to use the maximum of available information of both data sets for the first analysis of this kind performed in the present study, values in the regions of the other two masks (dense vegetation and snow cover/frozen ground) are considered in the analyses but will be discussed with care, recognizing the limitations of SM retrieval.




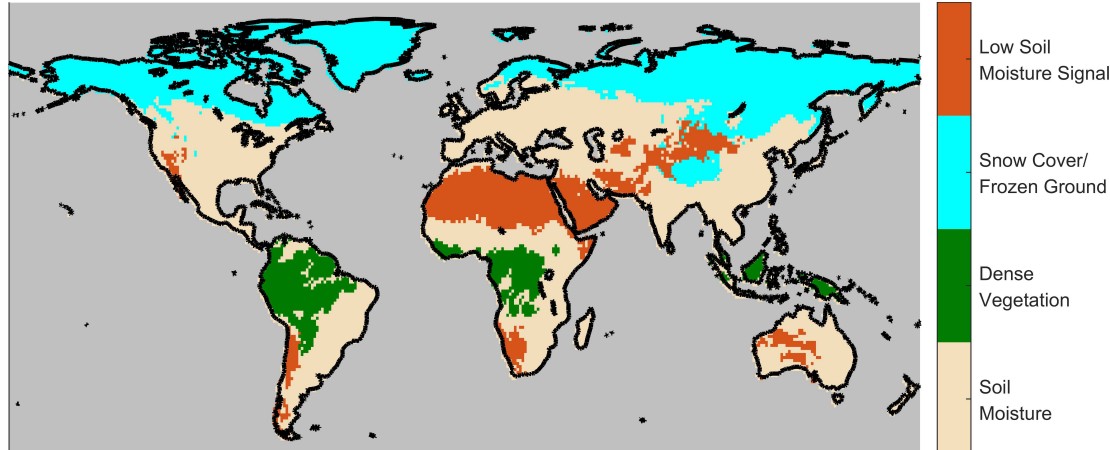

**Figure 2.** Spatial masks applied in this study for low SM variability (red), dense vegetation (green), snow cover and frozen ground (blue). In the other areas (beige, here denoted as "soil moisture"), reasonable satellite-based SM signals can be expected.

## 4 Results

### 4.1 Time series comparison for an example location

The comparison of time series between TWS from ITSG-Grace2018 and SM from various products is first shown for one exemplary grid cell. We chose one cell with marked short-term as well as seasonal SM variations, close to the city of Kota (25° N, 75° E), located in the south-east of the Indian state Rajasthan and characterized by a hot semi-arid climate with a monsoon season from July to September. Fig. 3 (top) shows daily TWS in this grid cell in comparison to SSM derived from satellite observations only. While TWS exhibits a comparatively smooth behaviour for short time scales and a dominant seasonal signal, the SSM time series are of considerably higher variability at short time scales. One reason is the noise of the satellite SM observations themselves (e.g., Karthikeyan et al. (2017)). On the other hand, this variability can partly represent a real signal as near-surface SM exhibits quick wetting and drying dynamics by individual precipitation events and by subsequent evaporation. In contrast, the much larger integration depth of the GRACE observations down to groundwater results in integral water storage with much slower dynamics. Furthermore, it has to be noted that for an integration depth of the SSM products of a few centimeters, the overall amplitudes of SM change in the order of 40 Vol% shown in Fig. 3 correspond to water storage changes that are by one order of magnitude smaller than those of GRACE-based TWS. An investigation whether the fast changing surface signals can also be detected in the GRACE data will be pursued in Section 4.3 for high-pass filtered time series. In a comparison of the three SSM products, SMAP L3 and the multi-satellite combination product ESA CCI time series appear less noisy than the SMOS L3 time series, which is in good agreement with findings of, e.g., Montzka et al. (2017), Cui et al. (2017), Xu and Frey (2021) and Kim et al. (2021).

While there is a general correspondence in the dynamics of TWS and SSM regarding the dominating seasonal signal (strong rise in water storage and SM during the monsoon season followed by a quick decline), a time shift can be identified between





both quantities. The seasonal maxima and minima of the GRACE time series occur approximately 1-2 months later than those of SSM. This can again be attributed to the slower and delayed water storage change in the deeper layers seen by GRACE, in which a change from dry to wet conditions (or vice versa) takes much longer to evolve than close to the surface. In addition to the main seasonal maximum, a minor secondary maximum can be identified each year in the period November to January in the SSM time series, particularly well visible in the SMAP L3 data (yellow line). For the TWS data, minor peaks or a less steep

TWS recession can be observed during this period, although a more general statement is hindered by the gaps in the TWS time series. Further discussion on this feature follows in the next paragraph.

Fig. 3 (middle) compares the TWS time series and the L4 SM products. For SMAP, both the surface (SSM) and the root zone (RZSM) product are the result of the data assimilation procedure described in Section 2.2.2. The SMAP L4 RZSM time series has a smaller variability compared to the SSM data both at short-term and at seasonal time scales. This is also the case for the

L4 RZSM product of SMOS, in which the seasonal variability is dampened even more strongly. The secondary maximum in Nov-Jan obvious in the L3 products is still visible in the L4 RZMS data set of SMOS since a mere extrapolation of the SSM data into deeper layers is applied for the SMOS L4 product. In contrast, the assimilation applied to SMAP L4 removes this second seasonal maximum, both for the SSM and the RZSM data sets. This indicates that the signal seen in the direct satellite SM data is not represented by the forcing data of the underlying model (i.e., precipitation) and thus fades out in the L4 SMAP

product. The secondary SM peak might thus be caused by extensive irrigation after the end of the monsoon season. However, a detailed analysis of this particular phenomenon is beyond the scope of this study.

The described characteristics of the different products are also confirmed when investigating the average year, i.e. the mean value for each day of the year shown in Fig. 3 (bottom). Even though the short overlapping time span does not allow for the computation of a stable climatology, the seasonal signal with SM increase during the monsoon phase between July and Septem-

ber can clearly be identified. The SSM products exhibit a larger seasonal amplitude than the dampened RZSM counterparts. Furthermore, the time-shift between SSM and TWS is generally larger than for the RZSM products. Here it can be concluded that the dynamics of RZSM are closer to the variability in the integral TWS signal that represents the entire water column, even though the RZSM data still misses the even slower processes in deeper soil layers and in groundwater.

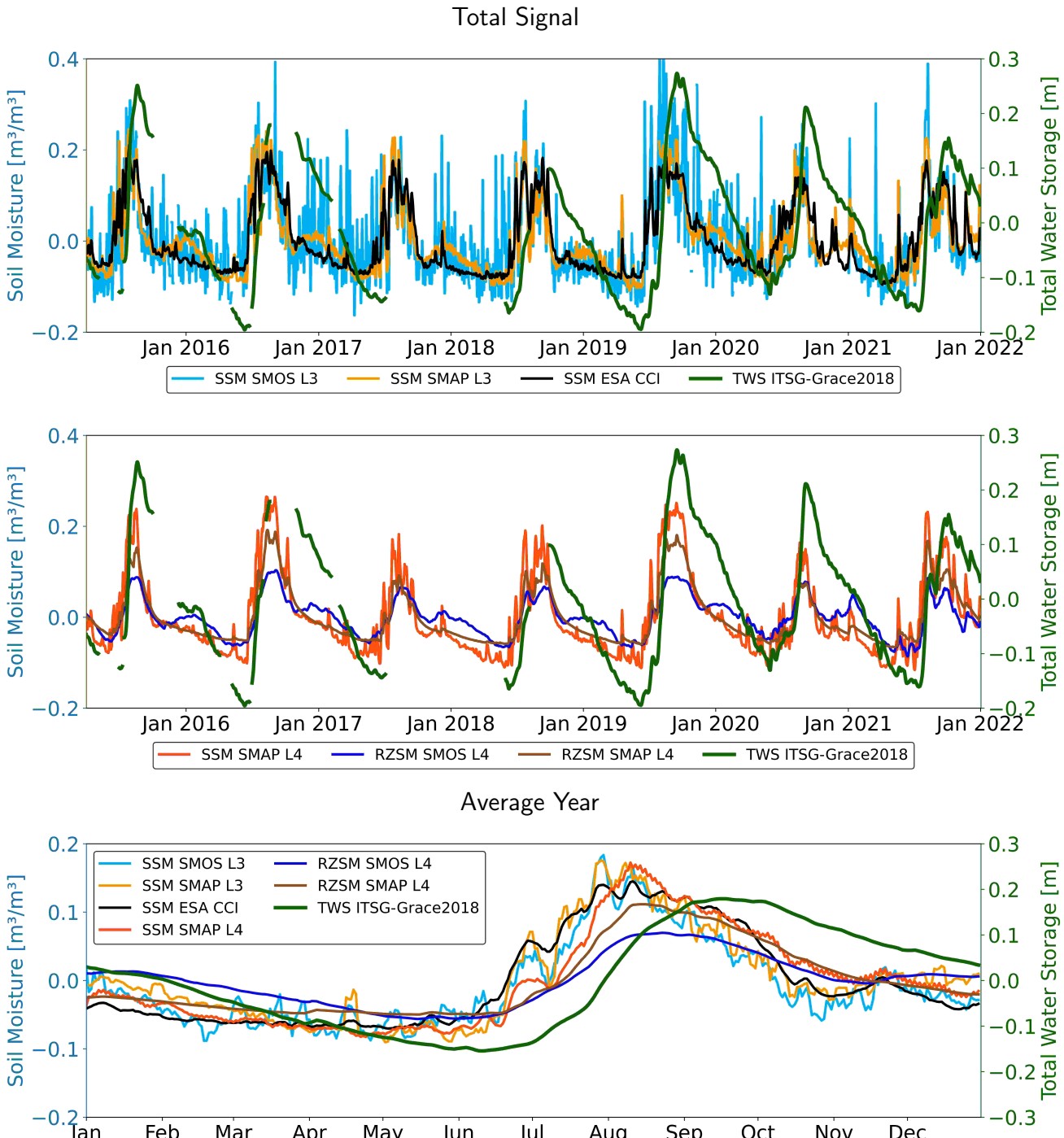

**Figure 3.** Time series of TWS from ITSG-Grace2018 and satellite SM for an exemplary grid cell around Kota, Rajasthan in India (25° N, 75° E). Top: TWS ITSG-Grace2018 vs SSM from SMOS L3, SMAP L3 and ESA CCI. Middle: TWS ITSG-Grace2018 vs SSM from SMAP L4 and RZSM from SMOS L4, SMAP L4. Bottom: Time series of the average year between TWS and all SM products.



## 4.2  Global analysis

The correspondence of the daily time series of TWS from ITSG-Grace2018 with the various SM products is analyzed globally
for each 1° continental grid cell and displayed as global maps of the correlation coefficient in Fig. 4. Desert areas according
to the definition provided in section 3.1 are excluded, as the GRACE signal is dominated by noise in these regions and no
reasonable comparison is possible. Additionally, Fig. 5 shows the histograms of cell-based correlation values, both globally as
well as separately for the major land cover types shown in Fig. 2.

The global patterns of regions with comparably large correlations between GRACE TWS and SM are similar for all SM
products. These regions are particularly in humid climate zones as well as in seasonally dry climates (parts of tropical South
America, south-east Asia, south-east U.S., northern Australia, outer tropics in Africa). The absolute correlation values markedly
differ between the SM products, though. The smallest correlations are found for the satellite-only L3 SSM data products
(Fig. 4a-c). While maximum values in individual grid cells can reach up to 0.92 (SMOS) and 0.93 (SMAP), the median values
amount to 0.23 (SMOS) and 0.27 (SMAP) only. SMOS L3 SSM has negative correlations (Fig. 5), particularly in the northern
latitudes that are at least partly influenced by snow cover and frozen ground (Fig. 5c), and less large positive correlations in
grid cells that are potentially well suited for SM remote sensing, i.e., that are not influenced by snow, ice or dense vegetation
(Fig. 5b). The latter might be attributed to a larger noise level in the SMOS than in the SMAP time series. The combination
data product ESA CCI (black curve in Fig. 5b) generally shows larger correlation values than both the single mission products
(except in the northern latitudes), particularly in South America, Africa and Australia (Fig. 4c). Please note that the masking
of, e.g., rain forest areas in the CCI product leads to an overall smaller number of grid cells for the comparison than for SMOS
or SMAP. The overall larger correlation of ESA CCI implies that the ensemble product down-weights spurious contributions of
individual data sets. Nevertheless, interpretation needs to be done with caution as TWS cannot directly be taken as a benchmark
for SSM.

The effect of using a land surface model with data assimilation for generating the SMAP L4 SSM product is a strong increase
in the correspondence of the temporal dynamics of this SM data set with GRACE-based TWS (Fig. 4d). The histogram in
Fig. 5a (red line) reveals most common correlation values in the range of 0.4-0.7, with a median of 0.52. This increase of
correlation for the L4 SSM data indicates that constraining the SM dynamics by a deterministic modelling approach and by
independently observed forcing data, such as precipitation rates and air temperature, removes a tangible part of the noise of the
L3 SSM product. For the L4 RZSM products of both SMOS and SMAP, their larger integration depth leads to another increase
in the agreement with TWS compared to the respective surface products (Fig. 4e+f). This again indicates that reflecting the
slower and delayed water transport processes in deeper soil layers in the L4 RZSM products causes their dynamics to be more
similar to the integrative GRACE-based TWS, in particular with respect to the temporal phase (see also the time shift analysis
below). This applies in particular for the grid cells that are reasonably suited for satellite-based SM monitoring for which the
histograms are shifted strongly to higher correlation values compared to the L3 data, with most common values of around 0.7
(SMOS) and 0.8 (SMAP) (Fig. 5b). In contrast, for SMOS L4 RZSM in high northern latitudes, correlations with TWS become
even more negative than for SMOS L3 SSM (see time shift analysis below). Already Xu et al. (2021a) show that the SMAP





L4 RZSM product outperforms the SMOS L4 product in terms of agreement with in-situ measurements from the International
Soil Moisture Network, particularly in the northern hemisphere.

**Figure 4.** Comparison of the correlation coefficient between all SM products and ITSG-Grace2018. Desert regions are masked out due to
low water variability.



**Figure 5.** Histograms of cell-based correlation values. (a): Showing the histogram of all grid cells globally. (b, c, d): Showing the grid cells for the major land cover types as defined in Fig. 2.

The different integration volumes and the different processes acting on SM and TWS cause a time shift between their respective dynamics (e.g., Fig. 3). We use cross-correlation analysis (see Appendix A1) for each grid cell to investigate the time shift on a global scale. On the resulting maps in Fig. 6, a negative time shift (blue colored grid cells) indicates a delay of the GRACE signal, i.e., the maximum of the dominant seasonal signal of TWS occurs later in the year than the maximum of SM. In the arctic regions mostly positive shifts (in some cases more than +120 days) are seen, in particular for the two





SMOS products (see also the histograms in Fig. 7). Snow accumulation leads to an increase and a maximum of TWS during the winter season whereas the maximum of SM is reached during the melting season several months later, resulting in a delay of SM versus TWS. This effect becomes even more pronounced when the further delay by SM storage in deeper layers is considered in the L4 root zone products of SMAP and in particular SMOS. Negative correlation coefficients of TWS and SM time series are a consequence of these inverse seasonal dynamics of the two storage terms (Fig. 5). Positive time shifts are also

seen in some tropical forest regions (e.g. Amazon, Congo), where the dense vegetation cover hampers the SM retrieval. This is particularly evident in the SSM L3 products (SMOS and SMAP) and persists in the SMOS RZSM, while the data assimilation introduced for SMOS L4 strongly reduces this effect. The positive time shifts can thus be attributed to artefacts in the SM data rather than to a hydrological signal.

In grid cells that are not influenced by neither snow cover nor dense vegetation (Fig. 7b), the time shifts are primarily

negative, indicating delayed dynamics of TWS compared to SM. For the majority of the products the most common values are in the range of -1 to -5 days, but time shifts can reach up to -90 days and beyond. A comparison of the different products reveals that the time shifts decrease (i.e. become less negative) the closer the SM products conceptually resemble TWS. The three surface SM products from ESA CCI, SMAP L3 and SMOS L3 show the highest negative time shifts (up to -60 to -90 days in Fig. 6) with median values of -35 days (both ESA CCI and SMAP L3) and -32 days (SMOS L3). The time shifts

are smaller for SMAP L4 (SSM)(median: -17 days) and even smaller for the RZSM products with median values of -5 days (SMAP L4 RZSM) and -13 days (SMOS L4 RZSM). This supports the assumption that adding SM dynamics of deeper soil layers to SSM increases the resemblance to the integrated TWS signal. For SMOS L4 SM, the extrapolation into deeper soil layers even leads to a change from a negative time shift to a positive time shift (i.e., TWS dynamics preceeding those of SMOS L4 ZRSM) in 17 % of the land areas, e.g., in parts of South America, Asia, and Australia. This points out possible deficiencies

of the depth-scaling algorithm for SMOS L4 in some regions, in a way that it represents transport and storage processes from the surface to deeper soil layers with too much delay and that the rates of reduction of water storage in the deeper layers by evapotranspiration and percolation and/or runoff may be underestimated, leaving too much water for a too long time in the storage.





Time-Shift between TWS (ITSG-Grace2018) and satellite soil moisture products

**Figure 6.** Time shift between SM and TWS time series per grid cell. Negative numbers imply that TWS is delayed in comparison to SM.

## Histograms of time-shifts

(a) All

(b) Soil Moisture

(c) Snow Cover/Frozen Ground

(d) Dense Vegetation

— SSM ESA CCI  — SSM SMOS L3  — SSM SMAP L3  — SSM SMAP L4  — RZSM SMOS L4  — RZSM SMAP L4

**Figure 7.** Histograms of cell-based time-shift. (a): Showing the histogram of all grid cells globally. (b, c, d): Showing the histograms for grid cells for the major land cover types which are shown in Fig. 2.

### 4.3 Sub-monthly variations (high-pass-filtered data)

The exemplary comparison of TWS and SM time series in Fig. 3 suggests that fast-changing SM signals might be masked in the TWS time series by the dominating slower dynamics of deeper layers in the unsaturated zone and in groundwater. Therefore, we isolate water storage changes on sub-monthly time scales by applying a high-pass filter (3rd order Butterworth filter) with





30-day cutoff frequency, see Appendix B for details. Exemplary high-pass filtered time series for 1.5 years of SMAP L4 RZSM and TWS from ITSG-Grace2018 are shown in Fig. 8 for the same grid cell as in Fig. 3. The overall correlation of the two time

series amounts to $\rho = 0.33$, but it varies strongly with signal strength. While in dry months (October to June) with very small high-frequency water storage fluctuations the GRACE time series is dominated by noise and only a very small correlation to RZSM of $\rho = 0.17$ was found, correlation is higher in the Monsoon season (July to September) with $\rho = 0.45$. Additional high-pass filtered time series for grid cells belonging to different climate zones are presented in the Appendix C.

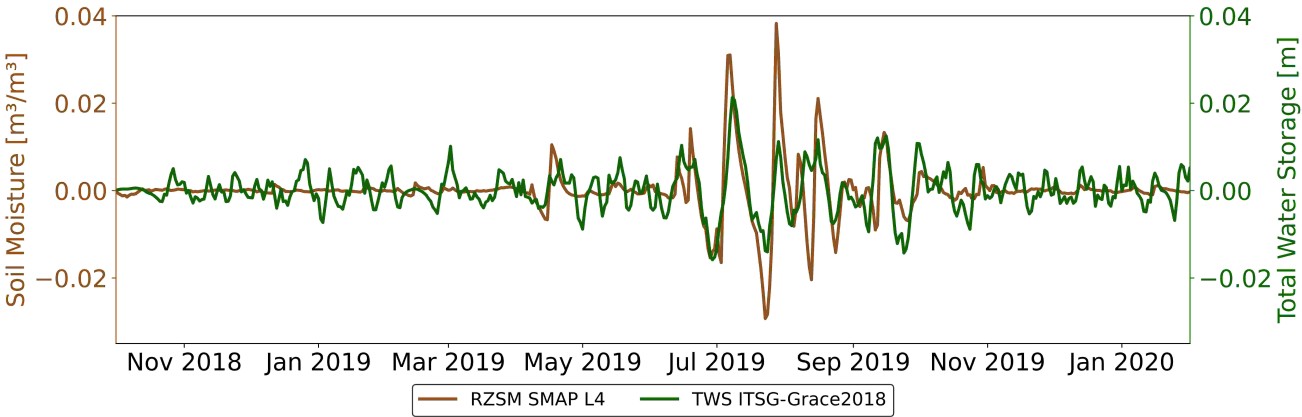

**Figure 8.** High-pass filtered time series of TWS from ITSG-Grace2018 and RZSM from SMAP L4 for an exemplary grid cell in India $(25^\circ$ N, $75^\circ$ E). Data gap between GRACE and GRACE-FO shortened.

The correlations of the high-pass filtered TWS and SM time series (Fig. 9) are considerably smaller than those of the

unfiltered time series. This can be expected as the seasonal storage variations that often cause high correlation values for the unfiltered time series are not present anymore in the filtered ones. Nevertheless, correlations of the sub-monthly signals are generally positive, with only a few grid cells showing negative correlations. Non-significant correlations are stippled in Fig. 9 (for more information on significance testing see Appendix A2). Again it can be observed that RZSM products have a stronger correlation with TWS than their SSM counterparts. While the numbers are very small for SSM of SMOS L3 (maximum value

of $\rho_{max} = 0.33$ and significantly positive correlations in only 26 % of land areas covered by the data product, desert areas excluded), they are larger for SMAP L3 SSM products ($\rho_{max} = 0.37$ and 43 % significantly positive). Again the larger noise floor of the SMOS data set most likely dominates the high-pass filtered time series. The share of grid cells with significantly positive correlations increases for the combination data product ESA CCI ($\rho_{max} = 0.38$, 58 % significantly positive) and even more for the data assimilated product SMAP L4 SSM ($\rho_{max} = 0.40$, 71 % significantly positive). The former reveals the

positive effect of combining several SSM data sets, which very likely results in a reduction of high-frequency noise, the latter indicates the influence of the forcing data of the underlying land surface model. In particular, it can be argued that model forcing with observation-based rainfall data causes major SSM increases to be closer in time and magnitude to water storage





increases that are seen by GRACE observations. For the root zone, the percentage of significantly positive grid cells is still low for SMOS L4 (37 %) with only a few larger areas in Western Brazil, in the south of Africa, and in India, but the magnitude of

the correlations has increased ($\rho_{max} = 0.50$). For SMAP L4 RZSM the values are significantly positive in the largest parts of the continents (77 %) with exceptions only in the very high latitudes and some small spots in dry regions in Africa, Australia, and Asia. In large regions (particularly in the south-eastern United States, in large parts of South America's south-east,in the Ganges-Brahmaputra basin and China, in North Australia) correlations are in the range of $0.5$, reaching maximum values of $\rho_{max} = 0.62$. Overall, the correspondence of the SM and TWS data sets presented here indicates that satellite gravimetry and

SM remote sensing are sensitive to the same hydrological dynamics even on time scales below one month.



## High-Pass Filter Correlation between TWS and satellite soil moisture products

(a)   SMOS L3 SSM  (b)   SMAP L3 SSM

(c)   ESA CCI SSM  (d)   SMAP L4 SSM

(e)   SMOS L4 RZSM  (f)   SMAP L4 RZSM

Correlation [-]

**Figure 9.** Correlation coefficients of high-pass filtered SM and TWS time series. Grid cells with non-significant correlations are stippled.

Finally, by computing the cross-correlation function for the high-pass filtered signals we determine for each grid cell the time shift of TWS versus SM time series that leads to the highest correlation (Fig. 10). Time shifts larger than $\pm 15$ days are masked out because they have no physical meaning given that a high-pass filter of 30 days was used. The remaining regions with valid results noticeably overlap with the regions where some correspondence of TWS and SM dynamics has already been

found in the correlation analysis with unshifted time series (Fig. 9). The time shifts found for the high-pass filtered time series are considerably smaller than those found for the full unfiltered signal. Time shifts tend to be negative with up to $-3$ days for





all SSM products (i.e., TWS is lagging behind SSM). For the RZSM products, however, the time shifts are less negative or close to zero (SMAP L4) or even positive (SMOS L4). The negative time shifts illustrate that GRACE observations represent the depth-integrated water storage dynamics in the subsurface that are delayed relative to SSM even at sub-monthly time scales.

This is corroborated by the observation that the time shifts mostly vanish when deeper layers are included in the SM product (SMAP L4), i.e., when the SM product becomes conceptually closer to the storage that is represented by the TWS observations. The positive time shifts of the SMOS L4 relative to TWS in most parts of the world indicate that the depth-scaling approach used for SMOS L4 may represent transport and storage processes from the surface to deeper soil layers with too much delay, similar to the results obtained with the full unfiltered signal. Even for the short time scales considered here with the high-pass

filtered signal, some processes such as daily evapotranspiration or runoff that cause water to be removed from the storage may be underestimated.

Time-Shift HPF between TWS and satellite soil moisture products

(a) SMOS L3 SSM        (b) SMAP L3 SSM

(c) ESA CCI SSM        (d) SMAP L4 SSM

(e) SMOS L4 RZSM        (f) SMAP L4 RZSM

≤-5   -4   -3   -2   -1   0   1   2   3   4   ≥-5

Time-Shift [days]

**Figure 10.** Time shift between high-pass filtered soil moisture and total water storage time series per grid cell. Negative numbers imply that TWS is delayed in comparison to SM.

## 5   Conclusions and outlook

In this study we investigated the global relationship of satellite-based SM products and non-standard daily water storage observations from the GRACE and GRACE-FO satellite gravimetry missions. The SM products differ with respect to satellite data (SMAP, SMOS, or a combination of various satellites), soil depth (surface SM or root zone SM) and the degree of post-





processing (Level 3 or Level 4 data products). Comparisons were carried out by correlation analyses both for the full signal and for sub-monthly variations obtained via high-pass filtering the time series with a 30-days cutoff frequency. Strong correlations between TWS and the different SM products generally occur in the same regions. These regions are mainly characterized by a seasonally wet or semi-arid climate, such as the east and south of Africa, northern parts of India, east Australia, the south-

east of China, Eastern Europe, the north-west and south-east of the United States, and significant parts of South America's south-east. For many regions, TWS dynamics are delayed relative to the SM dynamics, both for short-term variations at the scale of few days as well as when considering the seasonal dynamics. In particular in cold and snow-dominated regions, low correlations between TWS and SM dynamics prevail and the seasonal TWS dynamics are ahead of the SM dynamics.

From a hydrological point of view, both quantities (SM and TWS) actually represent different quantities regarding their

spatial domain (first centimeters of soil (SSM) vs. root zone (RZSM) vs. integrated water column of all storage compartments (TWS)), their units (volumetric percentage of water in the soil vs. water mass), and their spatial and temporal variability. The observed (dis-)agreements of SM and TWS time series and the time shifts between them therefore give insights into the relationships between the different water storage compartments, including moisture variations in different soil depths. It should be noted though that in some regions non-hydrological effects that remain in the GRACE data (e.g., major earthquakes)

or storage variations in other terrestrial water storage compartments that have not explicitly been taken into account in the present study, in particular surface water bodies, might affect these relationships.

The fact that satellite gravimetry can detect high-frequency signals that are related to SM variations, at least in areas and time spans with sufficiently large short-term variability, is quite remarkable by itself and shown for the first time in this study. While standard products of the GRACE and GRACE-FO mission are monthly averages of water storage anomalies, the present

study adds a new thematic field where there is valuable information content in daily GRACE data beyond earlier examples, e.g. floods or hydro-meteorological fluxes (Gouweleeuw et al. (2018); Eicker et al. (2020)).

For regions in which SM plays a dominant role for TWS variations, the results indicate that satellite gravimetry can be used to identify differences between SM products. Hydrological processes that are relevant for redistributing water vertically in the soil, particularly the percolation into deeper soil layers, can be identified as time shifts between the SM and TWS time series.

In this respect, our results give a preliminary indication that gravity-based TWS variations might have the potential to assess different methods of depth scaling (i.e., methods that are used to extrapolate surface SM variations to deeper soil layers). Such an assessment might be based on the analysis of time shifts between TWS and the depth-scaled SM time series. Assuming that the TWS signal for the region of interest and for the relevant temporal scale is dominated by SM variations, the absence of a time shift between TWS and the depth-scaled SM time series might be considered as an indicator of a suitable depth-scaling

approach. For the analyses presented here, this tends to be the case for the SMAP Level 4 RZSM data. In contrast, negative time shifts (i.e., SM dynamics that are ahead of those of TWS) may indicate that the delay of depth-integrated SM dynamics introduced by soil the depth-scaling approach is not sufficient enough. In turn, positive time shifts after depth-scaling as found here for some regions for the SMOS Level 4 RZSM data may point out that the depth-scaling approach mimics soil water redistribution processes in a way that a too large delay of the SM dynamics is caused.





While in regions and time spans with large sub-seasonal storage variability satellite gravimetry can identify short-term SM changes, this does not work well in cases of low signal-to-noise ratios (SNR) of high-frequency variations. In the current study, spatial masking was applied so that desert areas were completely excluded from the analysis and some other regions (frozen ground, dense vegetation) were discussed separately from the areas that are not influenced by either effect. Nevertheless, there are still many areas worldwide where the SNR is high in certain periods, e.g., during the rain season, but low in other time

spans. Therefore, an additional temporal masking and the exploration of meta data such as snow flags or indicators of low SNR is promising for an extended comparison in future.

   Ongoing improvements of GRACE/-FO data processing, future improvements in gravity field determination with the GRACE Follow-On laser ranging instrument measurements and with next generation gravity missions (NGGM), such as a constellation of two GRACE-like missions operating simultaneously at differently inclined orbits (Purkhauser et al., 2020), give prospect

for an increase of temporal and spatial resolution of satellite-based TWS data in future. This can be expected to be particularly beneficial for analyzing fast-changing and rather small-scale SM variations. The present study provides first evidence of insights on hydrological dynamics that can be gained from such a combination of TWS and SM remote sensing data.

## Appendix A:  Metrics

### A1    Correlation and cross-correlation

Suitable metrics are required to compare the time series of soil moisture (SM) and terrestrial water storage (TWS) for each continental grid cell. Since a direct comparison of the absolute values of the two variables is not possible due to the different integration depths and units, we analyze their relationship using Pearson's pairwise correlation coefficient $\rho_{xy}$, which is defined as the covariance of two variables $(x, y)$ divided by the product of their standard deviations:

$$\rho_{xy} = \frac{\sum_{t=1}^{T}(x_t - \overline{x})(y_t - \overline{y})}{\sqrt{\sum_{t=1}^{T}(x_t - \overline{x})^2}\sqrt{\sum_{t=1}^{T}(y_t - \overline{y})^2}} \qquad (A1)$$

The summation is performed over all daily time steps $t$ of the available time series with a length of $T$ days.

   Possible time lags between TWS and SM time series are determined using cross-correlation analysis which identifies the time shift $k$ for which the two time series show maximum correlation. The concept of cross-correlation is shown in Eq. (A2) and Eq. (A3) according to Box et al. (1994) in which first the covariance $c_{xy}$ between the two time series $x_t$ and $y_t$ for a given time lag $k$ is calculated:

$$c_{xy}(k) = \begin{cases} \frac{1}{T}\sum_{t=1}^{T-k}(x_t - \overline{x})(y_{t+k} - \overline{y}) & \text{for k = 0,1,2,...n} \\ \frac{1}{T}\sum_{t=1}^{T+k}(y_t - \overline{y})(x_{t-k} - \overline{x}) & \text{for k = 0,-1,-2,...-n,} \end{cases} \qquad (A2)$$

where $\overline{x}$ and $\overline{y}$ denote the mean values of the time series and $t$ indicates the respective point in time. We use $n = 180$ in our analysis, resulting in a total of 360 different covariances for each grid cell ($k = -180$ to 180, $\pm 6$ months). From the covariance





the cross-correlation $r$ is computed as:

$$r_{xy}(k) = \frac{c_{xy}(k)}{s_x s_y} \tag{A3}$$

where $s_x$ and $s_y$ denote the standard deviations of the time series. The time lag between SM and TWS is the value of $k$ for which the maximum cross-correlations is obtained.

## A2   Significance test

To identify grid cells with significant correlations between different time series, we use hypothesis testing with the null hypothesis $H_0$ stating that the correlation is not significantly different from zero and the alternative hypothesis $H_A$ assuming a
non-zero correlation:

$$\begin{aligned} H_0 &: p = 0 \\ H_A &: p \neq 0 \end{aligned} \tag{A4}$$

The statistical test is carried out by computing the test variable $T$, which is distributed with $n-2$ degrees of freedom according to the Student's t-distribution:

$$T = \frac{|\rho| \sqrt{n-2}}{\sqrt{1-p^2}} \sim t_{n-2} \tag{A5}$$

In Eq. (A5), $n$ is the total number of days with both TWS and SM observations for the respective grid cell. The test assumes uncorrelated observations from one day to the next which, however, is not strictly the case for the time series at hand. An auto-correlation analysis was carried out for a large number of exemplary grid cells to determine a mean correlation length of 3 days for surface and 5 days for root zone products. Consequently, the degree of freedom was adjusted to $n/3-2$ (SSM) or $n/5-2$ (RZSM). For the significance test, we chose a significance level of $\alpha = 0.05$ and calculated the corresponding quantiles
$K = F_t^{-1}(\alpha, n/3-2)$ (SSM) or $K = F_t^{-1}(\alpha, n/5-2)$ (RZSM). If $T \leq K$, the null hypothesis cannot be rejected, and the correlation is assumed to be insignificant. If $T > K$, it is reasonable to assume that the alternative hypothesis is correct and that the correlation deviates considerably from zero.

## Appendix B:  Derivation of sub-monthly signals using a Butterworth high-pass filter

In this section, we demonstrate the computation of the high-pass filtered signals for the exemplary grid cell around Kota, India, which was also shown in Section 4.3 of the main text. Fig. B1 shows TWS from ITSG-Grace2018 (green lines) and RZSM from SMAP L4 (brown lines) over a 15-month period (October 2018 to December 2019). The Butterworth high-pass filter with a cutoff frequency of 30 days was applied to remove signals with periods longer than 30 days and thus to isolate sub-monthly fluctuations. The darker green and brown colours illustrate the high-pass filtered signal for both variables. Additionally, the
corresponding low-pass filtered signals (i.e. containing signals with periods longer than 30 days) are added as dotted lines.





Summing up the high-pass filtered signal and the low-pass filtered signal again results in the original full signal (shown in the respective brighter colours). It can clearly be seen that the high-pass filtered time series capture the fast variations present in the original time series. These time series are the same as those displayed in Fig. 8 of the main text.

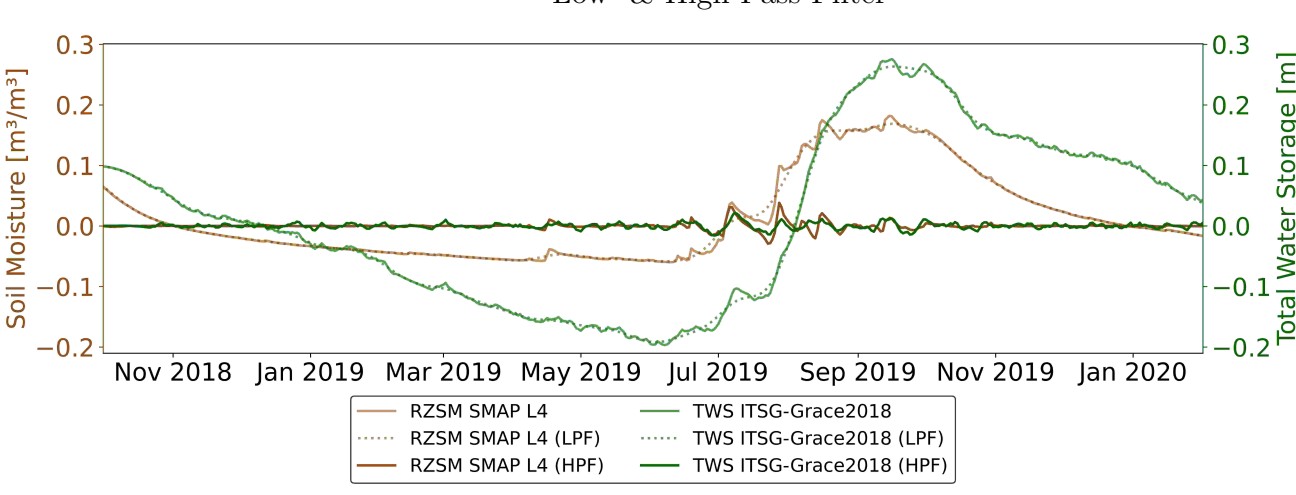

**Figure B1.** Sub-monthly signals using a Butterworth high-pass filter (HPF), together with the full and the low-pass filtered (LPF) time series, here shown for TWS from GRACE and root zone soil moisture (RZSM) from SMAP

## Appendix C: Other examples of high-pass filtered time series

In addition to the high-pass filtered signal computed for the example grid cell in Kota, India, as shown before, Fig. C1 displays the high-pass filtered signals for SM from SMAP L4 RZSM and TWS from ITSG-Grace2018 in some other geographical locations of different aridity. These places were chosen to illustrate the dynamics of the two variables under different environmental conditions and signal characteristics. Fig. C1a is for a location in the northwest of the state Arkansas in the United States. The climate is humid sub-tropical with a hot summer and no specific dry season. Accordingly, the time series, show

fast fluctuations and high correspondence of SM and TWS throughout the year, with a correlation of $\rho = 0.43$. At a location in Hungary (Fig. C1b) with a humid continental climate with warm summers, rain can fall at every time of year, while snow can fall during the winter months. The high-pass filtered SM and TWS time series show a correlation of $\rho = 0.31$ at this location. The time series of Fig. C1c are for a location north of the Kalahari Desert in Botswana. The climate there is semi-arid and mostly dry throughout the year, but there is a wet period with strong rainfall events during the summer. These events are clearly

visible in both TWS and SM time series with good correspondence, while no correlation and largely noise in the TWS data is visible during the dry period. At this location, the correlation over the entire time period is low, with $\rho = 0.11$, but it is higher if only the precipitation period was considered, with $\rho = 0.23$.





**Figure C1.** High-pass filtered time series for grid cells in different climatic conditions



*Data availability.* The daily GRACE data products of ITSG-Grace2018 are publicly available from TU Graz (https://www.tugraz.at/institute/ifg/downloads/gravity-field-models/itsg-grace2018/). The ESA CCI soil moisture product is can be downloaded from ESA's Climate Change Initiative (CCI) web-

page (https://esa-soilmoisture-cci.org/). Soil moisture products (Level 3 & Level 4) from ESA's SMOS mission are made available by the Centre Aval de Traitement des Données SMOS (CATDS) (https://www.catds.fr/Products/Available-products-from-CPDC). Soil moisture products (Level 3 & Level 4) from NASA's SMAP mission are provided by the National Snow and Ice Data Center (NSIDC) (https://nsidc.org/data/smap/smap-data.html).

*Author contributions.* Conceptualization: D.B., A.E.; Methodology: D.B., A.E., L.J., Data Curation: D.B.; Software: D.B., L.J.; Investi-

gation: All authors; Visualisation: D.B.; Funding Acquisition: A.E.; Writing - original draft: All authors; Writing - review ... editing: All authors.

*Competing interests.* The authors declare that they have no known competing financial interests or personal relationships that could have appeared to influence the work reported in this paper.





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
