# Peer review of "A global analysis of water storage variations from remotely sensed soil moisture and daily satellite gravimetry"

_Hydrology and Earth System Sciences, 2022_

## Author Comment (AC1)

**Reviewer #1:**

We would like to thank you very much for your helpful and constructive review. We write our responses in blue underneath each of your comments.

Best regards,

Daniel Blank (on behalf of all co-authors)

OVERVIEW

The study investigates the relationship between soil moisture and satellite gravimetry total water storage variations at daily scale and on a global scale. Multiple soil moisture products have been analysed, both for the surface layer and the root zone. The correlation and the time shift among satellite gravimetry total water storage and soil moisture products have been investigated in depth.

GENERAL COMMENTS

The paper is well written and clear. The investigation of daily terrestrial water storage (TWS) variations from GRACE(-FO) has been carried out only in a very limited number of studies and hence their global analysis is surely of interest for the readership of Hydrology and Earth System Sciences. However, I have found some major comments that needs to be addressed carefully.

- MAJOR: In the analysis of different soil moisture products, any disagreement with TWS variations from GRACE is attributed to error in soil moisture product. For instance, when SMOS L4 product has positive time shift with GRACE it is attributed to errors in the algorithm for obtaining root zone soil moisture from SMOS, but it might be an error on GRACE (of course, particularly at daily temporal resolution). Instead of only identifying the area of disagreement, a more detailed discussion should be carried out to shed light on the potential causes for that.

Thank you very much for this comment. Yes, of course it is completely correct that disagreement is not only caused by errors in soil moisture products, but can also be attributed to limitations in the GRACE TWS data set. We will, therefore, extent the discussion of limitations of GRACE, e.g. with respect to the limited temporal and spatial resolution of the TWS data, the noise floor of the daily time series, and the issue of signal separation (full vertically integrated water column including surface water bodies etc.). For this purpose, we will add more information directly in Section 2.1 (description of GRACE data set) and extent the discussion of the results. Nevertheless, we also want to point out that one hypothesis for this study is that the soil moisture products and GRACE TWS do show differences that are not only due to respective errors in both data sets, but that are physically based as the data sets represent different quantities. To explore what we might learn from such differences on hydrological process dynamics is one motivation for this study.

- MAJOR: The investigation with SMAP L4 product should be carried out separately from the other products. SMAP L4 is mostly a modelled product, the contribution of SMAP data is quite limited as highlighted in the analysis for the pixel in India. All satellite soil moisture products are able to identify the irrigation signal, whereas it is not the case for SMAP L4 as it is mostly modelled and it does not include the irrigation component. A paper clearly showing this aspect is going to be published soon. I believe

the analysis with SMAP L4 should be likely removed, or considered completely separately (note that many other modelled products can be considered as well).

While we do agree that for the SMAP L4 data sets it should be clearly stated that they are largely model based and satellite soil moisture is only used for data assimilation, we would very much like to keep them in the study. To assess the information content and possible value of the daily GRACE TWS data, we argue that most might be learned when comparing them to a range of soil moisture products with different signal characteristics. Officially, SMAP L4 is still denoted as a SMAP, i.e., remote sensing, data product. Nevertheless, in the revised manuscript we will stress more strongly, that these data sets are heavily influenced by the data assimilation and thus by the climate data used as input for the land surface model (e.g. precipitation), to leave no doubt that they are not purely based on satellite surface soil moisture observations.

- MINOR: Throughout the paper many acronyms are present without the definition, please add.

Thank you very much for pointing this out! We have added the explanations of all acronyms at their first appearance in the manuscript.

- MODERATE: At line 192 it reads that the linear trend is removed. I believe it would be very interesting to compare the products in terms of their long-term trends. Can the authors add this analysis?

We agree that the analysis of the trends in two different observation types is an interesting investigation on its own, which we would like to leave for a future study. In the present study, we focus on the information content of the daily GRACE TWS data at the sub-yearly and sub-monthly time scale, showing for the first time their signals related to soil moisture dynamics, which we consider to be of sufficient scope for this paper.

In the specific comments I have added several suggestions to improve the manuscript.

SPECIFIC COMMENT (L: line or lines)

L39: Soil moisture can be obtained from microwave but also optical data. If GNSS is mentioned, also optical data should be.

Thank you very much, we will add this to the text.

L53-55: Currently, well established approaches have been exploited for estimating root zone soil moisture from satellite surface soil moisture data. For instance, the operational service under Copernicus providing the Soil Water Index and the EUMETSAT H SAF root-zone soil moisture products. These products should be mentioned and I believe the sentence should be revised.

We will revise the sentence accordingly and add the reference to the mentioned Copernicus data sets.

Figure 2: I would change the text in the legend for "soil moisture". For instance, "committed area", or something similar.

We agree that the label „soil moisture" was not the best choice. We have, therefore, decided to change it to „area suited for analysis".

L241: These evaluations are valid for the analysed pixel, it should be clear. It seems to read general results.

We will adjust the sentence.

L256: It's not merely extrapolation, there's a physically approach for getting root-zone soil moisture from surface data.

Thank you! Yes, we will correct the sentence to make this more clear.

L260-261: It clearly shows that SMAP L4 is a modelled product not including irrigation, should be considered apart.

Yes, the results corroborate the deficiency of the SMAP L4 product, i.e., that it is mainly driven by precipitation input in the data assimilation framework and thus does not represent the impact of irrigation on soil moisture. Interestingly, with our study this result can not only be backed by purely observation-based near-surface soil moisture products but also by daily TWS observations based on satellite gravimetry.

Figure 3: In the figures the anomalies are shown. It should be clarified in the y-axis labels.

The figure will be adjusted accordingly.

L288: Exactly, GRACE TWS cannot be considered as a reference.

We will make this more clear also in the general discussion of the results (see also our comment above).

L294-295: SMAP L4 does not remove the noise, it is simply a modelled product.

Together with adjusting the text on the SMAP L4 data (as mentioned above), we will also clarify this point.

L329-333: Deficiencies might be due also in GRACE data, right?

As mentioned above, we will extent the discussion of the limitations of the GRACE data and mention here that the deficiencies can be caused by both data products, besides by physically-based differences between the two data types because they do not represent the same hydrological quantities.

Figure 7: It is not readable, please improve.

We will update the figure. Thanks a lot!

Figure 8: In the caption it reads "Data gap between …" Not clear, please revise.

We will update the caption. Thanks a lot!

L364-365: It seems to me the authors are overselling the results, the correlations in the high-pass filtered signal are very low. Only relatively better with SMAP L4, but it's not a satellite-based product.

We would like to point out that thorough significance testing including the consideration of temporal correlations has been carried out. It revealed significant correlations even for the high-pass filtered time series. Nevertheless, following the reviewer's comment, we will carefully check and eventually adapt the discussion to avoid possible overselling.

Figure 9: The range of the colorbar should be reduces. Otherwise the figure provides little information.

We will update the figure. Thanks a lot!

---

## Author Comment (AC2)

**Reviewer #2:**

We would like to thank you very much for your helpful and constructive review. We write our responses in blue underneath each of your comments.

Best regards,

Daniel Blank (on behalf of all co-authors)

The paper explores the linkage between SM and TWS datasets at global scale and daily timestep (by focusing on correlation and temporal shifts among the considered datasets) in order to provide new insights on sub-surface hydrological processes. The topic is relevant and well suited to HESS and the paper is clear, concise and well written. Here below, please find relevant comments.

Thank you very much for this positive feedback!

Main issues:

(*) what are the main limitations and future perspectives of this application? According to this, the final discussion should be expanded, by focusing on e.g. TWS data limitations (in terms of reliability) and human influence.

We will extend the discussion of limitations of TWS, e.g. with respect to the limited spatial and spatial resolution of the TWS data, the noise floor of the daily time series, and the issue of signal separation (full vertically integrated water column including surface water bodies etc.). For this purpose, we will add more information directly in Section 2.1 (description of GRACE data set) and extent the discussion of the results. Notwithstanding these limitations, the perspectives that are opened by the comparison of soil moisture and TWS data in terms of a better understanding of subsurface water transport dynamics and assessing depth scaling approaches of near-surface soil moisture observations as outlined in the conclusions chapter remain valid.

(*) How is it possible to deal with human influence on TWS and its relationship with SM dynamics? Is this a limitation?

While we are not fully sure on the context of the reviewer's question, we may stress that TWS based on satellite gravimetry as well as observation-based soil moisture capture the human influence on water storage, albeit with the differences that TWS is an integrative observation of water storage changes in all storage compartments, whereas soil moisture represents a subset only. Both observation-based-data sets may thus unravel deficits of modelling approaches that do not or not adequately represent such human influences and may contribute to model improvements. While both data products can do so for the effects of irrigation on near-surface soil moisture, for instance, GRACE-based TWS can provide additional information on the effect of such human impacts on water storage dynamics in the deeper unsaturated zone and in the groundwater, for the latter also on groundwater depletion eventually caused by water withdrawal for irrigation purposes.

(*) More details about the differences between L3 and L4 SM products should be provided (i.e. useful for readers not expert in SM)

Thank you very much for this suggestion. We will add more information on the conceptual differences between L3 and L4.

Minor issues

(*) all acronyms should be defined at first appearance

Thank you very much for pointing this out! We will update this in the text.

 (*)l. 219: remove "the" written twice

We will update the text. Thanks a lot!

 (*) Figure 2: improve figure resolution

We will update the figure. Thanks a lot!

(*) add a dot/symbol on the global map tp show where the case study grid cell is located

Yes, that is a very good suggestion. We will add it in the revised version of the manuscript.

---

## Author Response (AR1)

We would like to thank the editor, the two reviewers and one community commentator for their helpful und constructive feedback, which have helped us in improving the manuscript.

The most relevant changes to the mansucript include the following:

- We have extended the description of the limitations of the GRACE data sets and have included these limitations also in the discussion of the results.

- We have stressed the fact that SMAP Level 4 is a combined data set computed by data assimilation more clearly.

- We have adjusted some of the figures. In particular we substituted the too busy histogramms (as pointed out by Reviewer 1) by the corresponding cumulative distribution functions (CDFs) to show the statistics more clearly.

More detailed answers to each of the reviewer's comments can be found underneath each reviewer's comments below.

Best regards,

Daniel Blank (on behalf of all co-authors)

**Reviewer 1:**

OVERVIEW

The study investigates the relationship between soil moisture and satellite gravimetry total water storage variations at daily scale and on a global scale. Multiple soil moisture products have been analysed, both for the surface layer and the root zone. The correlation and the time shift among satellite gravimetry total water storage and soil moisture products have been investigated in depth.

GENERAL COMMENTS

The paper is well written and clear. The investigation of daily terrestrial water storage (TWS) variations from GRACE(-FO) has been carried out only in a very limited number of studies and hence their global analysis is surely of interest for the readership of Hydrology and Earth System Sciences. However, I have found some major comments that needs to be addressed carefully.

- MAJOR: In the analysis of different soil moisture products, any disagreement with TWS variations from GRACE is attributed to error in soil moisture product. For instance, when SMOS L4 product has positive time shift with GRACE it is attributed to errors in the algorithm for obtaining root zone soil moisture from SMOS, but it might be an error on GRACE (of course, particularly at daily temporal resolution). Instead of only identifying the area of disagreement, a more detailed discussion should be carried out to shed light on the potential causes for that.

Thank you very much for this comment. Yes, of course it is completely correct that disagreement is not only caused by errors in soil moisture products, but can also be attributed to limitations in the GRACE TWS data set. We will, therefore, extent the discussion of limitations of GRACE, e.g. with respect to the limited spatial and spatial resolution of the TWS data, the noise floor of the daily time series, and the issue of signal separation (full vertically integrated water column including surface water bodies etc.). For this purpose, we will add more information directly in Section 2.1 (description of GRACE data set) and extent the discussion of the results. Nevertheless, we also want to point out that one hypothesis for this

study is that the soil moisture products and GRACE TWS do show differences that are not only due to respective errors in both data sets, but that are physically based as the data sets represent different quantities. To explore what we might learn from such differences on hydrological process dynamics is one motivation for this study.

- MAJOR: The investigation with SMAP L4 product should be carried out separately from the other products. SMAP L4 is mostly a modelled product, the contribution of SMAP data is quite limited as highlighted in the analysis for the pixel in India. All satellite soil moisture products are able to identify the irrigation signal, whereas it is not the case for SMAP L4 as it is mostly modelled and it does not include the irrigation component. A paper clearly showing this aspect is going to be published soon. I believe the analysis with SMAP L4 should be likely removed, or considered completely separately (note that many other modelled products can be considered as well).

While we do agree that for the SMAP L4 data sets it should be clearly stated that they are largely model based and satellite soil moisture is only used for data assimilation, we would very much like to keep them in the study. To assess the information content and possible value of the daily GRACE TWS data, we argue that most might be learned when comparing them to a range of soil moisture products with different signal characteristics. Officially, SMAP L4 is still denoted as a SMAP, i.e., remote-sensing, data product. Nevertheless, in the revised manuscript we will stress more strongly, that these data sets are heavily influenced by the data assimilation and thus by the climate data used as input for the land surface model (e.g. precipitation), to leave no doubt that they are not purely based on satellite surface soil moisture observations.

- MINOR: Throughout the paper many acronyms are present without the definition, please add.

Thank you very much for pointing this out! We have added the explanations of all acronyms at their first appearance in the manuscript.

- MODERATE: At line 192 it reads that the linear trend is removed. I believe it would be very interesting to compare the products in terms of their long-term trends. Can the authors add this analysis?

We agree that the anaysis of the trends in two different observation types is an interesting investigation on its own, which we would like to leave for a future study. In the present study, we focus on the information content of the daily GRACE TWS data at the sub-yearly and sub-monthly time scale, showing for the first time their signals related to soil moisture dynamics, which we consider to be of sufficient scope for this paper.

In the specific comments I have added several suggestions to improve the manuscript.

SPECIFIC COMMENT (L: line or lines)

L39: Soil moisture can be obtained from microwave but also optical data. If GNSS is mentioned, also optical data should be.

Thank you very much, we have added this to the text.

L53-55: Currently, well established approaches have been exploited for estimating root zone soil moisture from satellite surface soil moisture data. For instance, the operational service under Copernicus providing the Soil Water Index and the EUMETSAT H SAF root-zone soil moisture products. These products should be mentioned and I believe the sentence should be revised.

We have revised the sentence acccordingly and added the reference to the mentioned Copernicus data sets.

Figure 2: I would change the text in the legend for "soil moisture". For instance, "committed area", or something similar.

We agree that the label „soil moisture" was not the best choice. We have, therefore, decided to change it to „area suited for analysis".

L241: These evaluations are valid for the analysed pixel, it should be clear. It seems to read general results.

We have adjusted the sentence.

L256: It's not merely extrapolation, there's a physically approach for getting root-zone soil moisture from surface data.

Thank you! We have changed the sentence.

L260-261: It clearly shows that SMAP L4 is a modelled product not including irrigation, should be considered apart.

Yes, the results corroborate the deficiency of the SMAP L4 product, i.e., that it is mainly driven by precipiation input in the data assimilation framwork and thus does not represent the impact of irrigation on soil moisture. We have added a sentence stressing the fact that here the strong influence of the model on the SMAP L4 data is clearly shown. Interestingly, with our study this result cannot only be backed by purely observation-based near-surface soil moisture products but also by daily TWS observations based on satellite gravimetry.

Figure 3: In the figures the anomalies are shown. It should be clarified in the y-axis labels.

The figure has been adjusted accordingly.

L288: Exactly, GRACE TWS cannot be considered as a reference.

We have made more clear at this point and also in the general discussion of the results (see also our comment above).

L294-295: SMAP L4 does not remove the noise, it is simply a modelled product.

Together with adjusting the text on the SMAP L4 data (as mentioned above), we have also clarified this point and re-written the sentence accodingly.

L329-333: Deficiencies might be due also in GRACE data, right?

In general, you are of course very right to point out that deficiencies in the compraison can always be caused by both data products, besides by physically-based differences between the two data types because they do not represent the same hydrological quantities. In the revised version we have made this point more clear throughout the text. However, we cannot think of a plausible physical reasoning that could cause events to show up in the GRACE time series „too early". Despite all known challenges involved with GRACE data (a description of which we have now included in Section 2.1), such spurious positive time shifts have not been reported, yet. We have extended the discussion on this matter at the mentioned point in the text.

Figure 7: It is not readable, please improve.

Thanks a lot. We have exchanged the very busy histogramms by the corresponding cumulative distribution functions (CDFs), that show the distribution of the pixels more clearly.

Figure 8: In the caption it reads "Data gap between …" Not clear, please revise.

We have changed the sentence.

L364-365: It seems to me the authors are overselling the results, the correlations in the high-pass filtered signal are very low. Only relatively better with SMAP L4, but it's not a satellite-based product.

Following the reviewer's comment, we have adapted the sentence and toned down the results. However, we would like to point out that thorough significance testing including the consideration of temporal correlations has been carried out. It revealed significant correlations even for the high-pass filtered time series.

Figure 9: The range of the colorbar should be reduces. Otherwise the figure provides little information.

We have updated the figure. Thanks a lot!

**Reviewer 2:**

The paper explores the linkage between SM and TWS datasets at global scale and daily timestep (by focusing on correlation and temporal shifts among the considered datasets) in order to provide new insights on sub-surface hydrological processes. The topic is relevant and well suited to HESS and the paper is clear, concise and well written. Here below, please find relevant comments.

Thank you very much for this positive feedback!

Main issues:

(\*) what are the main limitations and future perspectives of this application? According to this, the final discussion should be expanded, by focusing on e.g. TWS data limitations (in terms of reliability) and human influence.

We will extend the discussion of limitations of TWS, e.g. with respect to the limited spatial and spatial resolution of the TWS data, the noise floor of the daily time series, and the issue of signal separation (full vertically integrated water column including surface water bodies etc.). For this purpose, we will add more information directly in Section 2.1 (description of GRACE data set) and extent the discussion of the results. Notwithstanding these limitations, the perspectives that are opened by the comparison of soil moisture and TWS data in terms of a better understanding of subsurface water transport dynamics and assessing depth scaling approaches of near-surface soil moisture observations as outlined in the conclusions chapter remain valid.

(\*) How is it possible to deal with human influence on TWS and its relationship with SM dynamics? Is this a limitation?

While we are not fully sure on the context of the reviewer's question, we may stress that TWS based on satellite gravimetry as well as observation-based soil moisture capture the human influence on water storage, albeit with the differences that TWS is an integrative observation of water storage changes in all storage compartments, whereas soil moisture represents a subset only. Both observation-based-data sets may thus unravel deficits of modelling approaches that do not or not adequately represent such human influences and may contribute to model improvements. While both data products can do so for the effects of irrigation on near-surface soil moisture, for instance, GRACE-based TWS can provide additional information on the effect of such human impacts on water storage dynamics in the deeper unsaturated zone and in the groundwater, for the latter also on groundwater depelation eventually caused by water withdrawal for irrigation purposes.

(\*) More details about the differences between L3 and L4 SM products should be provided (i.e. useful for readers not expert in SM)

Thank you very much for this suggestion. We have added some more information on the conceptual differences between L3 and L4.

Minor issues

(\*) all acronyms should be defined at first appearance

Thank you very much for pointing this out! We have added the explanations of all acronyms at their first appearance in the manuscript.

 (\*)l. 219: remove "the" written twice

We have changed the sentence.

 (\*) Figure 2: improve figure resolution

We have improved the figure resolution.

 (*) add a dot/symbol on the global map tp show where the case study grid cell is located

Yes, that is a very good suggestion. We have added it in the revised version of the manuscript.

**Community Comment:**

I read through this interesting manuscript focused on assessing relationships between soil moisture (SM) and GRACE-based daily TWSA on a global scale. Thanks to the authors for this great contribution to the literature. I have two comments/suggestions that the authors may find relevant while revising their manuscript.

Thank you very much for contributing to the discussion of our manuscript and the positive and constructive feedback.

Lines 62-64. Daily TWSA has also been successfully employed to analyze the development and propagation of the water extremes using standardized drought and flood potential index (SDFPI). Please see Xiong et al. 2022a.

We have added the reference to the water extremes (drought/flood index) to our text.

More importantly,
Lines 388-389: From these lines, I understood that climate is (as portrayed herein) the major factor for a strong correlation between TWS and SM. [Line 401: not only surface water bodies but also human activities such as groundwater extraction in north India can affect these relationships significantly]. In my understanding, the larger the groundwater extraction for irrigation, the more positive will be the trends in SM, hence the more declining trends in TWS [due to the eventual loss of irrigated GW as runoff, evapotranspiration, and atmospheric moisture content]. Please see Xiong et al., 2022b (third paragraph of section 3.2). How do the authors relate the effect of such human-induced activities to their analysis? Additionally, how this human-related part (e.g., irrigation) is reflected in various SM products as we go deeper.
Overall, I could not find a sufficient description of human activities in the manuscript (though partly touched upon in line 260), which I think should be accommodated, at least as the explicit uncertainty discussion in the analysis and/or future research directions.

Similar to our response to the question of Reviewer 2 on the human influence above, we may stress that TWS based on satellite gravimetry as well as observation-based soil moisture capture the human influence on water storage, albeit with the differences that TWS is an integrative observation of water storage changes in all storage compartments, whereas soil moisture represents a subset only. While both observation-based soil moisture data and GRACE-based TWS observations can do so for the effects of irrigation on near-surface soil moisture, GRACE-based TWS observations can provide additional information on the effect of such human impacts on water storage dynamics in the deeper unsaturated zone and in the groundwater. While we agree with the comment above that groundwater extraction for irrigation purposes may be seen as a long-term TWS decline in the GRACE data, we doubt that this will necessarily lead to a positive trend in soil moisture as there are presumably high losses by ET and drainage, as also mentioned in the comment above. The combination of (near-surface) soil moisture products and TWS observations may shed light on how human acitivtes and irrigation practices in particular translate into water storage changes in deeper soil zones. We want to stress, though, that the focus of this study was on shorter-term storage changes and not on long-term trends. We will nevertheless add a comment in the uncertainty discussion of the revised manuscripts on overlapping trend signals in the unsaturated zone and in the groundwater that are difficult to disentangle in the TWS data.

**References**

Xiong et al., 2022a. A Novel Standardized Drought and Flood Potential Index Based on Reconstructed Daily GRACE Data. Journal of Hydrometeorology. https://doi.org/10.1175/JHM-D-22-0011.1

Xiong et al. 2022b. Leveraging machine learning methods to quantify 50 years of dwindling groundwater in India. Science of the Total Environment. https://doi.org/10.1016/j.scitotenv.2022.155474